# From Health-in-All-Policies to Climate-in-All-Policies: Using the Synergies between Health Promotion and Climate Protection to Take Action

**DOI:** 10.3390/ijerph21010110

**Published:** 2024-01-18

**Authors:** K. Viktoria Stein, Thomas E. Dorner

**Affiliations:** 1Karl-Landsteiner Institute for Health Promotion Research, 3062 Kirchstetten, Austria; gesundheitsfoerderung@karl-landsteiner.at; 2Department for Public Health and Primary Care, Leiden University Medical Centre, 2333 The Hague, The Netherlands; 3Academy for Ageing Research, Haus der Barmherzigkeit, 1060 Vienna, Austria; 4Centre for Public Health, Department for Social and Preventive Medicine, Medical University Vienna, 1090 Vienna, Austria

**Keywords:** one health, health promotion, climate change, exercise, nutrition, health literacy, climate literacy, sustainability, social determinants

## Abstract

The climate crisis is developing into a life-changing event on a global level. Health promotion with the aim to increase the health status of individuals, independent of the present health status, has been developed on a scientific basis at least for the last eight decades. There are some basic principles which are prerequisites for both health promotion and climate protection. Those principles include (1) sustainability, (2) orientation on determinants, and (3) requirement of individual as well as community approaches. People are generally aiming to protect their lifestyle habits (e.g., traveling and consumer habits) and personal property (e.g., car and house) with easy solutions and as little effort as possible, and this can affect both health and climate. To reduce the emission of greenhouse gases and to protect our environment, changes towards a sustainable lifestyle have to be embedded into everybody’s mind. Examples for domains that need to be addressed in health promotion as well as in climate protection include (health and climate) literacy, physical activity and active mobility, and nutrition and dietary habits. If health promotion fails to tackle those domains, this will continue to drive the climate crisis. And climate change, in turn, will affect health. On the other hand, developing and promoting health resources in the domains mentioned could help to mitigate the health-damaging effects of climate change. Success in the joint efforts to promote health and protect the climate would improve the One Health approach, the health of people and the environment.

## 1. Introduction

The impact of climate change on health has recently been acknowledged in various international publications and calls for urgent action issued by the COP26 [1], the WHO [2,3], and the Lancet Commission [4], among many others. The link was also made explicit in the UN’s Sustainable Development Goals (SDGs) [5], especially goals 2, 3, 6, 13, and 15.

The WHO now calls climate change the single biggest health threat facing humanity with environmental factors taking the lives of around 13 million people annually. Over 90% of people are exposed to unhealthy levels of air pollution, largely resulting from burning fossil fuels, which in turn further drives climate change. In 2018, air pollution from fossil fuels caused USD 2.9 trillion in health and economic costs, about USD 8 billion a day. The direct damage costs to health are estimated to be between USD 2 and 4 billion per year by 2030. [6] The impact of droughts and the subsequent damages to agriculture will increase considerably, with an estimated 40% of the global population living in areas under severe water stress by 2050 and approximately five million more deaths will be attributable to climate change [7]. Increasingly, it is also acknowledged that rising temperatures will have a real impact on productivity and job performance. The International Labour Organisation [8] forecasts that the equivalent productivity of 80 million full-time jobs will be lost by 2030 due to heat-related stress. These are only some of the examples of how climate change now impacts every aspect of individual and community health. Given these multi-faceted and multi-layered effects of climate change on the health and well-being of individuals and communities, it is becoming ever more urgent to think about ways in which synergies can be found to build resilient individuals, communities, and systems, which can enable a climate-protective and health-promotive lifestyle.

This review paper intends to add to the ongoing discussion by building on existing strategies, frameworks, and concepts, thus bringing a novel perspective to the table. Instead of focusing on how climate change impacts health, we suggest to bring together well-established concepts of health promotion and community involvement to identify practical solutions, which would both improve health and protect the climate. This paper is a review of existing literature and is aimed at researchers to incite interdisciplinary thinking and research into the field of health promotion and climate protection. It also calls for researchers, practitioners, and policy makers to benefit from the obvious synergies, which a joint understanding of health promotion and climate protection can yield, in an age where resources are scarce and a lot more has to be accomplished in both fields to ensure a healthy, equitable, and safe future. This article wants to inspire researchers to look beyond their respective fields and answer some of the most pressing research questions of our age, including how can we implement sustainable solutions which benefit both health and climate? Which actions have the most impact on health and climate? How can the existing concepts and frameworks be translated into practice?

## 2. The Relationship between Health Promotion and Climate Protection

The connection between health promotion and climate protection has most recently been brought together by the WHO in its global framework for integrating well-being into public health utilising a health promotion approach [9]. The global framework gives guidance to policymakers on how to implement the six strategic directions, first defined by the Geneva Charter in 2021 [3]. This charter in itself updated the principles of health promotion set out by the WHO’s Ottawa Charter [10] and aligned them with the SDGs [5] and the concept of planetary health [7]. The six strategic directions outlined in the global strategy address the need to adopt a planetary health approach to foster healthy environments, to continue the efforts of implementing universal health coverage by strengthening primary health care, health promotion, and community empowerment, and to ensure sustainable change through a rigorous approach to measuring and monitoring progress.

As outlined in the introduction, the impact of climate change on individual and collective health is well established [1,2,4,6,11,12]. The WHO identifies the creation and maintenance of healthy environments as a core element of primary prevention [2]. In the WHO’s global strategy on health, environment, and climate change (2020) [13], the authors describe the various connections between drivers for climate change and drivers for ill health, e.g., how poorly planned urban environments and unsustainable traffic infrastructure create noise and heat islands, increase air pollution, reduce the possibilities for physical activity, and limit green spaces. Limited green spaces in themselves increase temperatures and reduce carbon monoxide absorption but also adversely impact mental health and physical activity. A similar negative feedback loop can be observed with air pollution [6,13].

Based on work in Project Drawdown, Mailloux et al. (2021) [12] proposed nine fields of climate action, which would also promote human health: improved air quality, increased physical activity, healthier diets, reduced risk of infectious disease, improved sexual and reproductive health, and universal education. While the connections are obvious, there is an ongoing call for more research and evidence into the interrelationship between climate action and public health [2,9,13]. The following sections offer a basis for discussion on how to analyse these interrelationships using existing concepts to create a truly interdisciplinary research approach.

## 3. Breaking Down the Complexity—Three Principles to Protect the Climate and Promote Health

There are some basic concepts which are prerequisites for both health promotion and climate protection. The need to promote sustainable change on all levels, from the individual to the system, necessitates participatory approaches, which involve individuals and communities in co-designing and implementing solutions. A clear understanding of the determinants for health, well-being, and climate protection strengthens these solutions.

(1)Sustainability

Both climate protection and health promotion can only be successful when actions, initiatives, campaigns, and policies are geared towards long-term, sustainable change in habits, behaviours, and lifestyles. In its groundbreaking “Special report on climate change and health: the health argument for climate action”, the COP26 summarised the evidence on how health-promoting policies and lifestyles can also benefit a healthy climate [1]. For example, policies and actions to facilitate walking and cycling as climate-friendly means of transport improves health through increased physical activity, resulting in reductions in respiratory diseases, cardiovascular diseases, some cancers, diabetes mellitus, and obesity [14]. Similarly, the promotion of urban green spaces reduces exposure to air pollution, generates local cooling effects, helps to relief stress, and increases recreational space for social interaction and physical activity [15,16]. A shift to more nutritious plant-based diets in line with the WHO’s recommendations could reduce global emissions significantly, ensure a more resilient food system, and avoid up to 5.1 million diet-related deaths a year by 2050 [17]. But all of these actions require sustainable changes from individuals and communities.

In connection with climate protection, the definition of sustainability as drawn up by the World Commission on Environment and Development (Brundtland Commission) is used very frequently. This definition is, “meeting the needs of the present without compromising the ability of future generations to meet their own needs” [18]. In relation to health, many different definitions of sustainability are used in the scientific literature. Most of these definitions revolve around the fact that measures are still taken after a certain period of time, that changed lifestyle habits are maintained for a longer period of time, or that health effects can still be proven after a certain period of time [19]. There is a certain tradition in health promotion of organising and financing this in projects with a defined beginning and a defined end. However, sustainability can only be guaranteed in non-terminated programmes, which is why there has been a long debate about health promotion in projects. A large number of factors have been identified that either support or hinder the sustainability of health promotion programmes [20]. Many models have been introduced to support long-term lifestyle changes and maintain behaviour modification; one of the best known being the Trans-Theoretical Model, which describes the different stages of change [21]. The strength of this model lies in the fact that it allows not only for progress, but also for relapses into old habits, which are not depicted as failures, but as a natural occurrence in individual change behaviour: sustainable change is not achieved through linear progress, but through a start, maintain, fail, and try again process, which can spiral upwards or downwards. Similar approaches need to be developed for behaviour changes for climate protection, and the lessons learned from decades of health promotion should be taken into account.

Using the obvious synergies of programmes to promote healthier lifestyles and at the same time to help protect the climate, may also widen the audience of individuals and communities that can be reached. A comparison of the Social Ecological Model of Health [22] with the SDGs [5] illustrates this interrelationship and strengthens the argument for concerted efforts in the area of health promotion and climate protection (see Figure 1).

(2)Focus on determinants

The Social Ecological Model of Health (SEM) [22] describes the multi-faceted determinants of health on all levels, from the individual to the system. It builds on the realisation that 90% of one’s health outcomes are influenced by factors other than access to clinical care [23,24]. Similarly, the Sustainable Development Goals (SDGs) are an evolution of the Millenium Development Goals (MDGs) recognising that health, economic and social development and our natural environment are inextricably intertwined. Table 1 highlights the alignment between the determinants of health as defined by the SEM and the SDGs, but it is best-captured in the Ottawa Charter’s statement that “Health is created and lived by people within the settings of their everyday life; where they learn, work, play, and love” [10].

Social determinants of health (SDH) are among the best-researched and -verified factors that facilitate or hinder the development of health [25]. Social capital, for example, represents the relationships between people and their community, and these resources are associated with the ability and motivation of individuals for adaptations towards specific topics as well as their overall well-being [26]. Increasing individual social capital and increasing social cohesion on a community level is an important part of individual and public health promotion. It is also part of SDGs 11 and 16.

Likewise, climate protection is dependent on social determinants, and projects worldwide have shown that social cohesion in society favours changes towards a sustainable and climate-friendly economy. Data of rural households in Ethiopia were investigated to identify correlations between social capital and climate change adaptation [27]. Similar results of high social capital and climate change adaptation in the agricultural sector were published by investigating data of Indonesian farmers, who had to deal with an increasing risk of pest attacks. A strong sense of solidarity and belonging to the community engaged 70% of the farmers to contribute to adaptation processes to solve issues responsible for crop failures [28].

The SEM and SDH concepts and evidence of their validity have been around for decades, and so has their description of the influence of the living environment on health. Nevertheless, the impact of the environment and climate on health has only emerged as a priority area for action through the recent discussions on climate change itself, the SDGs, and recent publications from the WHO [6]. Again, this is an example where research and practice on climate protection can learn a lot from the research on health promotion, and the climate aspect on the determinants of health should be strengthened. The example of the WHO’s Healthy Cities network already illustrates how the two concepts are interlinked with the overall aims of a Healthy City including the creation of a health-supportive environment and the provision of basic sanitation and hygiene needs (https://www.who.int/activities/creating-healthy-cities, accessed on 16 January 2024).

(3)Individual as well as community approaches

While health promotion is often seen as taking individual action, climate protection is perceived as a collective and global challenge. However, for sustainable and positive change to happen, both for health promotion and climate protection, a combination of individual and community approaches is necessary. Individual approaches include, for example, sustainable behaviour changes towards a healthier lifestyle and the reduction in the CO_2_ footprint by switching to more sustainable modes of transport, like cycling or public transport. Another key element is for people to be aware of how daily choices influence the environment and can contribute towards climate protection, just as their lifestyle choices affect their health. Health literacy forms one pillar of the WHO’s definition of health promotion [29], and in parallel, climate literacy needs to be established as a key element of climate protection.

Individual and community approaches therefore need to apply participatory strategies, which enable and support people to actively co-design, decide, and implement solutions to fit into their context and life. Change is always difficult and will only be sustainable if people understand why it is necessary, how it will affect them, and how it will fit into their living environment and daily life. This is true for both health promotion [9,22] and climate protection measures. A key term used in both arenas is empowerment. In health, empowerment refers to “the process by which people develop their intrinsic capabilities to increase control over the factors, decisions and actions that affect their health and care and the process of gaining power externally over them” [30]. Tools to implement empowerment exist on the individual and community levels and include strengthening health literacy, enabling informed choice and the co-production of solutions. There is a plethora of examples on how individuals can be more actively involved in their health and care, and these are usually linked to the concept of person-centred health systems [31,32]. In addition to the involvement and empowerment of individuals, the health arena is increasingly working together with communities to create healthy environments and contextualise service delivery according to the needs of the local population. Concepts like population health management emphasise the importance of proactive health promotion and prevention over reactive treatment and management of diseases [33,34]. Similar principles are used in health promotion, for which the key strategies of health promotion are advocacy, mediation, and empowerment [10,29]. This again reflects the need to recognise and enforce the rights and responsibilities of individuals and communities, to collaborate across sectors and interest groups, and to recognise communities as equal partners in the design and implementation of solutions. In parallel, the Action for Climate Empowerment (ACE) adopted by the UN Framework Convention on Climate Change wants to empower all members of society to engage in climate action through the six ACE elements—climate change education and public awareness, training, public participation, public access to information, and international cooperation on these issues (https://unfccc.int/ace#Article-6-of-the-Convention, accessed 16 January 2024).

While health promotion recognises the need to use both individual as well as community approaches [10,22,29], in the Paris Agreement, the call for action is issued mainly to institutions and systems, including governments, cities, regions, businesses, and investors. The lack of individual and community involvement or empowerment in the official climate policies and strategies has found its counterpart in the direct action taken by global movements like Fridays for Future or the Last Generation. It would strengthen climate action if similar individual and community approaches were used to harness the power and expertise of local communities.

## 4. Examples for Common Domains of Health Promotion and Climate Protection

There is a multitude of domains that affect both health and climate. Examples discussed here are (1) literacy, (2) physical activity, and (3) nutrition and dietary habits. Some of them will be challenged by the burden of climate conditions (e.g., physical activity, diet, and food supply), and others will be more important than before (e.g., training and education of the public and health professionals, fostering social cohesion, and building social capital). If health promotion fails to tackle those domains, this will drive the climate crisis and consequently yield a variety of health issues. On the other hand, developing and promoting health resources in the mentioned domains could help to mitigate the health-damaging effects of climate change [35].

(1)Literacy

According to an OECD definition, literacy refers to reading and writing skills in a broader sense. It is the ability to recognise, understand, interpret, create, communicate, and evaluate written materials in different contexts. According to this definition, literacy enables persons to achieve their goals, develop their knowledge and potential, and fully participate in society [36].

The WHO defines health literacy as the achievement of “a level of knowledge, personal skills and confidence to take action to improve personal and community health by changing personal lifestyles and living conditions” [37]. This definition implies that health literacy means more than the ability to obtain access to, read, and understand health information (functional health literacy). It also creates the foundation for which persons are empowered to play an active role in improving their own health by critically evaluating the value of health information (critical health literacy), applying the information and being an equal player in the healthcare system (interactive health literacy), and lobbying for better health conditions in the society. Health literacy is more and more understood as a social determinant of health [38], and improving health literacy in the community needs both interventions aimed at individuals and interventions that create the conditions for this at the population level [39].

Climate literacy can be understood as the ability to know and understand the principles of climate, impacts of climate change, and approaches to adaptation or mitigation [40]. By analogy with the definitions of literacy and health literacy, climate literacy would mean more than just reading and understanding. Individuals should be able to critically evaluate and apply the information and challenge decision makers and lobby for sustainable policies and changes at the societal level. The basic principles of salutogenesis, as defined by Antonovsky as dimensions of the sense of coherence [41], can also be applied to the genesis of climate protection. Comprehensibility, a trust that connections with regard to climate change are understandable and that future climate developments are predictable, is an important prerequisite. Manageability means the trust that climate protection measures not only can be actually be implemented, but also do not necessarily mean significant economic and personal loss of rights. Meaningfulness means the trust that things in life like climate protection are interesting, worthwhile, and a source of satisfaction. It also means the ability to see measures of climate protection in a larger context, also beyond the borders of one’s own generation and present time.

To create literacy, both health and climate literacy needs campaigns and trainings by local, national, and global stakeholders (e.g., policy and public health) and a framework for education (e.g., health and climate literacy in schools, climate change curriculums in medical and health professional education, and health aspects in ecological education curriculums) [42,43]. Limaye et al. summarised the results of 72 peer-reviewed articles on climate change, human health, and education or training and thereby established a framework for climate and health literacy [43]. They grouped the identified elements into three categories: (1) functional literacy, including knowledge about causalities and mechanisms, how factors worsen both the climate crisis and human health, and how climate change affects human health; (2) intermediate literacy, including understanding of determinants and intertwining of environment and human health, implications of climate changes, and the possible impact of interventions towards climate protection; and (3) advanced literacy, including understanding the evidence derived from different types of data and models as well as understanding the complexity and variations over space and time [43]. Clear parallels can be discerned with the Action for Climate Engagement (ACE).

(2)Physical activity

Physical activity (PA) is widely used for the prevention, therapy, and rehabilitation of chronic diseases. Also, PA increases well-being and quality of life and therefore is a well-established tool in health promotion [44]. PA increases physical fitness and thereby creates resilience against many non-communicable diseases including health burdens as consequences of the climate change-related heat episode, like heat strokes and circulatory problems [45,46]. This connection is contrasted by the fact that older and vulnerable groups in particular might experience a possible negative health impact if PA is carried out in the heat. However, these dangers can be mitigated if special safety measures are followed on hot days.

Changing and maintaining the lifestyle towards more PA is a major challenge. In order to reach sustainability in this regard, orientation towards the determinants, including the social determinants, of PA behaviour is a basic requirement. In addition, sustainable implementation of this health behaviour requires individual measures aimed at improved health literacy and the acquisition of skills to carry out health-enhancing PA and physical training, as well as society-, public-, and community-based measures that create the conditions for this, such as a change towards a culture in which PA is considered socially desirable, and creating physical, environmental, structural, and legislative requirements for safe PA.

PA not only has positive consequences for health, but also for the climate. Especially transport-related PA (e.g., walking, running, cycling, riding an e-bike, and riding a scooter) to cover daily distances and thereby replace motorised vehicles that consume fossil fuels are discussed to be associated with a lower production of greenhouse gases and therefore contribute towards climate protection.

Green spaces are important areas for climate protection due to their CO_2_-binding capacity. Additionally, parks and green spaces play an important role to avoid so-called urban heat islands [15,16,47]. Furthermore, green spaces are prerequisites and important determinants of PA behaviour. In fact, a study in Austria has shown a strong correlation between green spaces and the perceived accessibility of green spaces with the amount of PA on a local level [48]. 

PA is expected to change because of climate change. Climate change influences the weather towards higher temperatures, more heat days, higher humidity, and more extreme weather events. This is expected to influence physical activity behaviour. The amount of PA is reduced in both cold and hot temperatures, as well as on precipitation days [49]. As hot days become more frequent due to climate change, it is to be expected that PA will decrease in many countries for a longer period of time in the summer. Another aspect of climate change is the decreased duration and accumulation of snowfall, especially in mountainous areas. This affects winter sports and as a consequence decreases the amount of PA activity in winter. In fact, a study from an Austrian Alpine region showed that the intention to participate in recreational winter sports activities decreased due to climate change [50]. People with a higher socio-economic position might find it easier to regularly participate in PA, disregarding the weather outside. The geographical region is also important, because in colder regions, climate change might increase the amount of PA and in wormer regions, climate change might decrease the amount of PA.

According to the World Health Organization’s Global Action Plan on Physical Activity (GAPPA) 2018–2030, there are five domains related to PA, which are influenced by climate change. These are infrastructures, green spaces, exercise programmes, mass communication and mass participation events, and training of professionals. These domains are connected with climate policy, demonstrating clearly the link between strategies for physical activity promotion and the reduction in greenhouse gas emissions. To reach their goals, the GAPPA mainly focuses on four strategic objectives (active societies, active environments, active people, and active systems) and 20 policy actions and emphasises changes in PA policy making, proposing among other facts health economic assessments of health and climate and environmental benefits in the areas of active transport and urban design [51].

(3)Nutrition and dietary habits

The growing understanding of how our dietary choices impact not only our health but also the environment has led to a realisation that sustainable dietary practices can be a powerful tool for addressing both climate change and public health concerns. It must be acknowledged that not all climate-friendly foods are considered healthy and not all healthy foods are climate-neutral at the same time. However, plant-based diets, rich in fruits, vegetables, legumes, nuts, and whole grains are components of a healthy dietary pattern, and they have the potential to have lower carbon footprints compared to diets high in animal products. In the last decades, researchers provided data and evidence that sustainable nutrition in the frame of a sustainable lifestyle is beneficial for the environment as well as for the overall health and well-being of humans [52,53,54]. The One Health dietary approach, developed by the Food and Agriculture Organization of the United Nations [55], can contribute to both climate protection and health promotion.

Healthy diets are often associated with reduced risk factors for chronic diseases such as obesity, diabetes mellitus, cardiovascular diseases, and cancer. A shift towards plant-based diets can lead to lower saturated fat intake, reduced cholesterol levels, and improved weight management. Plant-based diets are rich in fibre, vitamins, and antioxidants which are linked to various health benefits [56,57,58].

Modern food systems, characterised by extensive livestock production, energy-intensive processing, and long supply chains contribute significantly to greenhouse gas emissions, deforestation, and biodiversity loss. In fact, diets and food quality have been identified to be responsible for 25% of global greenhouse gas emissions. Adopting sustainable diet patterns can mitigate these impacts [59,60,61].

Various determinants influence dietary choices, encompassing individual, cultural, economic, and social factors. Education, knowledge, and, finally, literacy play a vital role in promoting sustainable and healthy dietary behaviours. Furthermore, economic factors, such as the affordability and accessibility of healthy and sustainable foods, are crucial determinants. Not only individual choices, but also policies that support sustainable agriculture, subsidise nutrient-rich foods, and encourage local food production can create an enabling environment for healthier and more sustainable diets [62].

Climate change also affects food production. Agricultural productivity might decrease due to environmental degradation, desertification, soul depletion, overgrazing, rising sea levels, urban development, and increased usage of road for industrial use. Adverse weather events like flooding, landslides and erosion, storms, hurricanes, and droughts will further damage crops. Those mechanisms together with others like transport-related problems, spoilage and bacterial damage, and increased necessity for pesticide use will worsen nutritional problems associated with climate change [63]. All those factors contribute towards a loss of food diversity and a reduced supply of fresh foods such as vegetables and fruit and subsequently to an increase in the price of fresh and healthy food.

In Figure 2, we summarise the individually discussed factors and their connections. In the centre of the figure, you can see climate protection and health promotion, which influence each other. If climate protection fails, people’s health will worsen. On the other hand, health-promoting measures have the potential to contribute to climate protection at the same time. Examples of this include active mobility and a primarily plant-based diet, which both improve health and reduce CO_2_ consumption and thereby protect the climate. Appropriate literacy (climate and health literacy) makes appropriate lifestyle decisions substantially easier. Health promotion and climate protection are directly influenced by determinants, and both require individual and community approaches. Determinants and individual and community approaches are also the prerequisites for physical activity, nutrition, and literacy and thus indirectly influence climate protection and health promotion. Ultimately, sustainability in climate protection and health promotion is needed in order to actually achieve and maintain better health for people but also for the ecosystem, entirely in the spirit of One Health.

## 5. From Policy to Practice

There is now a long list of publications, which provide evidence on the impact of climate change on health and well-being, and an equally long list of international policies, frameworks, and strategies of how to tackle these challenges. What is still missing is the use of the collective experience in the health promotion and climate action arenas to create synergies and co-design and implement viable solutions with local communities and individuals to move towards true planetary health. Taking up the call of the WHO’s global framework [9] for whole-of-government and whole-of-society approaches “to generate health benefits for the populations and the planet”, these should be extended to include climate protection as a vehicle for health promotion and vice versa. Similar to the health-in-all-policies approach, a climate-in-all-policies approach could foster holistic solutions to complex challenges. Health and other experts need to be fully involved in climate decision-making processes at all levels to ensure health and equity considerations are well understood and accounted for when developing climate policies [1,2,3,4,9].

In Table 2, the synergies between health promotion and climate protection in the three mentioned examples are summarised, with a focus on the prerequisite for sustainability, determinants orientation, and individual and community approaches. 

## 6. Conclusions

Health promotion and climate protection do not overlap 100%, but they share many concepts and tools, which help to create synergies in tackling the challenges faced by societies and governments. Summarising these in three key principles of sustainability, focussing on determinants, and a combination of individual and public approaches organises the increasing evidence and the existing policies to outline clear areas for joint action. Understanding health promotion and climate protection as two interlinked fields of action would give researchers the opportunity to benefit from interdisciplinary collaboration and potentially demonstrate higher impact and societal value, which would in turn incentivise policy makers to invest in joint programmes. Using a societal and equity lens for this review also ensures that the calls of global movements and the SDGs to protect the rights of future generations are equally considered with current individual and collective concerns. A lot remains to be accomplished in order to build healthy, sustainable, and safe environments for people to thrive in, but many of the tools and solutions already exist. It remains for policy makers, researchers, and communities to come together and create sustainable change.

## Figures and Tables

**Figure 1 ijerph-21-00110-f001:**
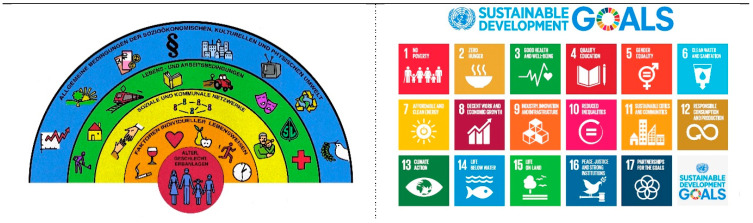
Social Ecological Model of Health and the Sustainable Development Goals. (**left**) Social Ecological Model of Health [19] (graphic: Fonds Gesundes Österreich); the headings in the semicircles from outside to inside are: “General socioeconomic, cultural and environmental conditions”, “Living and working conditions”, “Social and community networks”, “Individual lifestyle factors”, and “”Age, sex and constitutional factors”; (**right**) Sustainable Development Goals [5] (graphic: United Nations).

**Figure 2 ijerph-21-00110-f002:**
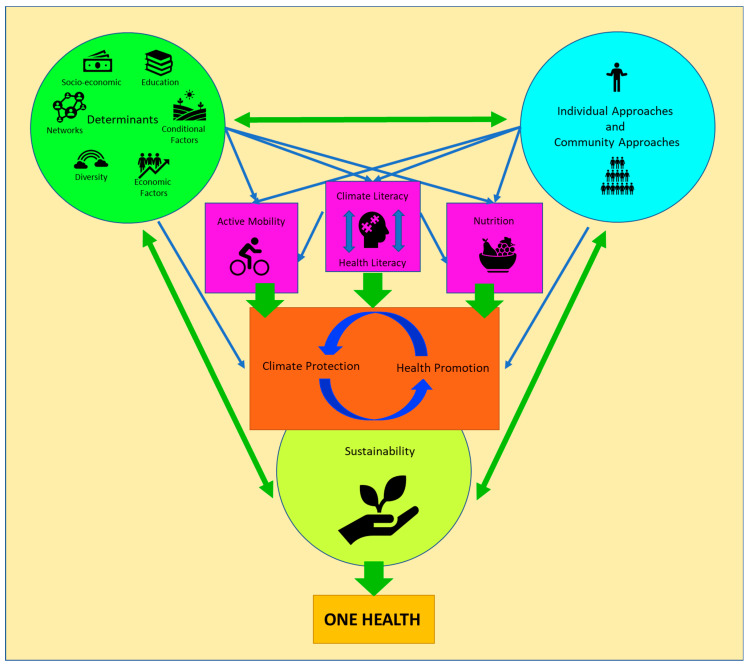
Relationships between health promotion and climate protection and their mutual prerequisites.

**Table 1 ijerph-21-00110-t001:** Alignment of the determinants of health with the SDGs.

SEM ^1^	SDG ^2^
Environmental conditions	Affordable and clean energy (7), sustainable cities and communities (11), climate action (13), life below water (14), and life on land (15)
Agriculture and food production	Zero hunger (2) and responsible consumption and production (12)
Education	Quality education (4)
Work environment	Decent work and economic growth (8)
Unemployment	No poverty (1)
Water and sanitation	Clean water and sanitation (6)
Health care services	Good health and well-being (3)
Social and community networks	Social and community networks
Sex	Gender equality (5)
Conditional factors	Industry innovation and infrastructure (9), reduced inequalities (10), and peace, justice, and strong institutions (16)

^1^ Social Ecological Model; ^2^ Sustainable Development Goals (number of the respective SDGs in brackets).

**Table 2 ijerph-21-00110-t002:** Recommendations for actions to use the synergies of health promotion and climate protection.

	Sustainability	Determinants	Individual and Community Approaches
**Literacy**	Adaptation of the dimensions of the sense of coherence, as a prerequisite for salutogenesis (comprehensibility, manageability, and meaningfulness) can also be seen as a prerequisite for sustainability in the genesis of climate protection.	Social determinants, especially education, play a key part in the levels of health and climate literacy. Easy access to evidence-based information in lay language is another important determinant. Facilitating these levers through active policies in education and health and climate promotion can mitigate negative impact of determinants.	Address individuals, patients, health professionals, and society as a whole to increase health literacy and climate literacy. Furthermore, the people, patients and public must be seen as important and equal partners for health promotion and climate protection. Including comprehensive education on health and climate from kindergarten, continuing to geriatric care, would establish a baseline understanding across society.
**Physical activity**	Bottom-up instead of top-down approaches and orientation on individual stages of change can promote a change in physical activity behaviour towards sustainable, active mobility.	The orientation of policies and strategies towards social, economic, cultural, individual, and health- and fitness-related determinants is an important prerequisite for the promotion of active mobility.	Encourage individuals towards physically active transportation as alternative for daily distances and create the conditions for it like green spaces and a traffic system which is safe and inviting for physically active transportation.
**Nutrition and dietary habits**	A sustainable change and maintenance of healthy eating habits is very difficult for many people. The best results can be achieved by considering all established pillars of health promotion (empowerment, participation, orientation towards determinants, personal needs and believes, and stages of change, etc.).	Social, economic, individual, and cultural determinants of eating habits must be taken into account for a healthy and climate-friendly change in diet.	Develop sustainable nutrition guidelines for healthy nutrition with focus on local, organic, and plant-based food and communicate them to individuals, stakeholders, and the public.

## Data Availability

No new data were created or analysed in this study. Data sharing is not applicable to this article.

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
