# Peer review of "From Health-in-All-Policies to Climate-in-All-Policies: Using the Synergies between Health Promotion and Climate Protection to Take Action"

_ijerph, 2024, doi:10.3390/ijerph21010110_

Round 1
Reviewer 1 Report (Previous Reviewer 1)
Comments and Suggestions for Authors
Am happy with the improvements made - it adds significantly to the clarity and hence value of the paper.
Comments on the Quality of English LanguageImproved and generally satisfactory.
Author Response
We want to thank the reviewer very much for all the comments!
Reviewer 2 Report (Previous Reviewer 4)
Comments and Suggestions for Authors
Thank you for incorporating the revisions; the paper has indeed seen improvement. However, there are still some minor adjustments needed before publication. The Introduction section appears brief and lacks a clear exposition of the paper's aim and approach. It is recommended to introduce certain information from Section 2 early in the introduction for better clarity. Specifically, pages 2, lines 87-103 should be referenced in the introduction. Section 2 needs to critically analyse the literature on both health promotion and climate protection to underscore the existing gaps and establish a more effective connection between them. In line 90, page 2, please provide clarification on the term "Novel Perspective" and elaborate on how its novelty has been determined.
Author Response
"Thank you for incorporating the revisions; the paper has indeed seen improvement."
We want to thank the reviewer very much for this positive feedback.
"However, there are still some minor adjustments needed before publication. The Introduction section appears brief and lacks a clear exposition of the paper's aim and approach. It is recommended to introduce certain information from Section 2 early in the introduction for better clarity. Specifically, pages 2, lines 87-103 should be referenced in the introduction."
We agree with the suggestion and have moved the lines 87-103 into the Introduction section.
"Section 2 needs to critically analyse the literature on both health promotion and climate protection to underscore the existing gaps and establish a more effective connection between them."
We agree with the reviewer. However, section 2 should be seen as more of a preparation for the further sections, in which the similarities between health promotion and climate protection are discussed in detail. In section 2 we tried to create a good compromise between detail and length of the entire manuscript. For better understanding, we have now better connected section 2 with the following sections.
"In line 90, page 2, please provide clarification on the term "Novel Perspective" and elaborate on how its novelty has been determined."
We thank the reivewer for the suggestion, we have now reformulated the sentence and hope this clarifies the novel perspective. See the text highlighted in yellow and here: "This review paper intends to add to the ongoing discussion by building on existing strategies, frameworks and concepts, thus bringing a novel perspective to the table. In-stead of focusing on how climate change impacts health, we suggest to bring together well-established concepts of health promotion and community involvement to identify practical solutions, which would both improve health and protect the climate."
This manuscript is a resubmission of an earlier submission. The following is a list of the peer review reports and author responses from that submission.
Round 1
Reviewer 1 Report
Comments and Suggestions for Authors
Not a scientific paper per se but a useful discussion and review of some of the major topics. The coverage is good and the arguments generally sound.
The English needs some correction and there are some odd words used, suggesting the need for a review for clarity.
Comments on the Quality of English LanguageThe English is generally good but needs some stylistic and grammar adjustment - see above. There are some oddities in the vocabulary.
Author Response
Dear reviewer, in the attachment you can find our reply to your comments.
Many thanks for your review of the manuscript.

Reviewer 2 Report
Comments and Suggestions for Authors
Well written paper focusing on alternate methods about creating a synergy between health promotion and climate protection. Need more scientific evidence about the association which i though was lacking in this paper. Have there been any meta-analysis or systematic reviews about impact of these.
Also would love to see more figures / pictures to create more impact to the audience in regards to impact of all the synergistic actions of health promotion and climate protection.
Author Response

(The authors gave the same response as above.)

Reviewer 3 Report
Comments and Suggestions for Authors
In this paper, the authors made a review of the principles of health promotion and climate protection.
The work is well documented, with enough bibliographic references, but for a better understanding, it should be rewritten, using more schemes.
Also, the abstract could be more specific. I suggest the authors should organize the abstract as well as the main text in four sections, namely: scope, objectives, methods, results, and conclusions.
The work is well documented, with enough bibliographic references, but for a better understanding, it should be rewritten, using more schemes.
Also, the conclusions are insufficient, they must be developed with the scoring of the main elements identified following the study of the specialized literature.
Author Response

(The authors gave the same response as above.)

Reviewer 4 Report
Comments and Suggestions for Authors
Thank you for submitting your intriguing paper. Unfortunately, there are substantial issues that need to be addressed before it can be considered for publication in this journal. Firstly, the primary objective of the paper is not readily apparent. It is essential to clarify the purpose of the paper, identify its key takeaways, and specify the intended audience. To facilitate a better understanding, authors should provide implementations to bridge the gap between the two areas under discussion. Clear research questions will help as well.
Furthermore, the literature review does not adequately establish the significance of the chosen topic. Both the paper and its abstract require thorough rewriting and restructuring to enhance the clarity of the research aim. Additionally, the paper's contribution to the existing body of literature needs to be better defined. A comprehensive review should extend beyond mere citation and paraphrasing, encompassing critical analysis, synthesis, and interpretation of prior research.
Indeed, it is imperative to engage in discussions that reflect the authors' perspectives. Moreover, the conclusion section, a crucial component of the article, is currently too brief. It should encompass clear policy implications, limitations, potential avenues for future research, and more.
Author Response

(The authors gave the same response as above.)
